# HPLC–PDA and LC–MS/MS Analysis for the Simultaneous Quantification of the 14 Marker Components in Sojadodamgangki-Tang

**Chang-Seob Seo** * 🅳 **and Mee-Young Lee**

Herbal Medicine Research Division, Korea Institute of Oriental Medicine, Daejeon 34054, Korea; cozy11@kiom.re.kr
* Correspondence: csseo0914@kiom.re.kr; Tel.: +82-42-868-9361

**Abstract:** Sojadodamgangki-tang (SDGT) is a traditional Korean medicine consisting of 12 medicinal herbs that has been used in Korea for the treatment of asthma since ancient times. However, the quality control of herbal formulas that contain two or more herbal medicines remains challenging. In this study, 14 marker components were analyzed simultaneously by using high-performance liquid chromatography with photodiode array detection in addition to the use of liquid chromatography–tandem mass spectrometry for quality evaluation of SDGT. The simultaneous determination of the 14 marker components was validated in terms of linearity, recovery, and precision. The established methods can provide useful data for the quality control of SDGT and related herbal formulas.

**Keywords:** HPLC–PDA; LC–MS/MS; simultaneous quantification; Sojadodamgangki-tang

## 1. Introduction

Traditional Chinese medicine (TCM), traditional Korean medicine (TKM), and kampo medicine (KM) have long been used in clinical practice for the treatment or prevention of various diseases in countries such as Korea, China, and Japan. The treatments are generally regarded as safe and have become standardized over centuries of refinement. However, with the development of modern science, research on these herbal prescriptions has become more frequent. Moreover, there is an increasing demand for regulatory documentation to certify the safety and standardization of TCM, TKM, and KM, and validated methods are therefore required.

Sojadodamgangki-tang (SDGT) is a TKM consisting of 12 herbal medicines that has been used to treat asthma [1]. The effects of SDGT on $SO_2$-induced respiratory injury, type I and type IV allergic reactions, and allergic asthma have been reported [2–5].

The main components in the medicinal herbs that make up SDGT are known to be as follows: phenylpropanoids (e.g., rosmarinic acid) and sesquiterpenoids (e.g., $\beta$-caryophyllene) from Perillae Fructus [6], phenols (e.g., homogentisic acid and 3,4-dihydroxybenzaldehyde) from Pinelliae Tuber [7], coumarins (e.g., decursin and decursinol angelate) from Angelicae Gigantis Radix [8], flavonoids (e.g., apigenin and schaftoside) from Arisaematis Rhizoma [9,10], flavonoids (e.g., hesperidin, nobiletin, and narirutin) from Citri Unshius Pericarpium [11,12], coumrins (e.g., praeruptorin A and B) from Peucedani Radix [13], lignans (e.g., honokiol and magnolol) from Magnoliae Cortex [14], flavonoids (e.g., poncirin and neoponcirin) from Ponciri Fructus Immaturus [15], triterpenoids (e.g., pachymic acid, dehydroeburicoic acid, and polyporenic acid C) from Poria Sclerotium [16,17], flavonoids (e.g., liquiritigenin), flavonoid glycoside (e.g., liquiritin apioside and liquiritin), and triterpenoid glycosides (e.g., glycyrrhizin) from Glycyrrhizae Radix et Rhizoma [18], phenols (e.g., 6- and 10-gingerol)

from Zingiberis Rhizoma Recens [19], and flavonoid glycosides (e.g., spinosin and 6‴-feruloylspinosin) from Zizyphi Fructus [20].

For the quality control of SDGT, we describe herein the development of the first simultaneous analysis of 14 marker components by high-performance liquid chromatography–photodiode array detection (HPLC–PDA) as well as liquid chromatography–mass spectrometry with tandem mass spectrometry (LC–MS/MS). The 14 marker components determined were: liquiritin apioside (LIQA), liquiritin (LIQ), and glycyrrhizin (GLY) (from Glycyrrhizae Radix et Rhizoma), nodakenin (NOD), decursin (DEC), and decursinol angelate (DECA) (from Angelicae Gigantis), narirutin (NAR) and hesperidin (HES) (from Citri Unshius Pericarpium), naringin (NARG), neoponcirin (NPON), and poncirin (PON) (from Ponciri Fructus Immaturus), apigenin (API) (from Arisaematis Rhizoma), honokiol (HON) (from Magnoliae Cortex), and praeruptorin A (PRAA) (from Peucedani Radix).

## 2. Materials and Methods

### 2.1. Plant Materials

The 12 component herbs of SDGT (Table S1) were purchased from the herbal medicine manufacturer Kwangmyungdag Medicinal Herbs (KMH, Ulsan, Korea) in November 2017. The origin of these raw materials was confirmed by herbalist, Seung Yeol Oh, CEO of KMH (Ulsan, Korea), based on "The Dispensatory on the Visual and Organoleptic Examination of Herbal Medicine" [21]. A voucher specimen (from 2018–CA01–1 to 2018–CA01–12) has been deposited at the Herbal Medicine Research Division, Korea Institute of Oriental Medicine (KIOM).

### 2.2. Chemicals and Reagents

The 14 reference standard compounds (Figure S1) used for the qualitative and quantitative analysis of SDGT were purchased from the following natural product suppliers: LIQA (CAS No.: 199796-12-8, 98.0%), PON (CAS No.: 14941-08-3, 98.9%), and PRAA (CAS No.: 73069-25-7, 99.6%) were purchased from Shanghai Sunny Biotech Co., Ltd. (Shanghai, China); LIQ (CAS No.: 551-15-5, 99.6%) and GLY (CAS No.: 1405-86-3, 99.0%) were purchased from Wako Pure Chemical Industries, Ltd. (Osaka, Japan); NOD (CAS No.: 495-31-8, 99.5%), NPON (CAS No.: 14259-47-3, 98.2%), and HON (CAS No.: 35354-74-6, 98.0%) were purchased from ChemFaces Biochemical Co., Ltd. (Wuhan, China); NAR (CAS No.: 14259-46-2, 99.5%), HES (CAS No.: 520-26-3, 98.6%), and API (CAS No.: 520-36-5, 98.8%) were purchased from Biopurify Phytochemicals Ltd. (Chengdu, China); NARG (CAS No.: 10236-47-2, 95.0%) was purchased from Merck KGaA (Darmstadt, Germany); DEC (CAS No.: 5928-25-6, 98.0%) and DECA (CAS No.: 130848-06-5, 98.0%) were purchased from NPC Bio Technology (Yeongi, Korea).

HPLC-grade reagents (methanol, acetonitrile, and water) and formic acid (FA, for HPLC) were purchased from J.T.Baker (Phillipsburg, NJ, USA) and Merck KGaA (Darmstadt, Germany), respectively.

### 2.3. Preparation of SDGT Water Extract

As demonstrated in Table 1, the 12 crude herbs that make up SDGT were combined, 50 L of distilled water was added, and the mixture was extracted under pressure (98 kPa) at 100 °C for 2 h using an electric extractor (COSMOS-660; Kyungseo E&P, Incheon, Korea). The extracted SDGT aqueous solution was filtered using a standard sieve (53 μm mesh) and then lyophilized to obtain 635.3 g (yield, 12.71%) of powder extract.

**Table 1.** Linear range, regression equation, $r^2$, limit of detection (LODs), and limit of quantification (LOQs) for marker compounds using high-performance liquid chromatography–photodiode array detection (HPLC–PDA) ($n$ = 3).

| Compound | Linear Range (µg/mL) | Regression Equation [a] $y=ax+b$ | $r^2$ | LOD (µg/mL) | LOQ (µg/mL) |
|---|---|---|---|---|---|
| LIQA | 1.56–100.00 | $y = 14{,}516.82x + 1576.68$ | 0.99999 | 0.04 | 0.13 |
| LIQ | 0.78–50.00 | $y = 20{,}297.12x + 832.37$ | 0.99999 | 0.07 | 0.20 |
| NOD | 1.56–100.00 | $y = 32{,}616.18x - 1073.11$ | 0.99998 | 0.33 | 1.01 |
| NAR | 1.56–100.00 | $y = 20{,}454.28x + 2443.73$ | 0.99999 | 0.08 | 0.25 |
| NARG | 2.34–150.00 | $y = 16{,}038.76x + 4074.11$ | 0.99996 | 0.37 | 1.11 |
| HES | 2.34–150.00 | $y = 17{,}321.70x + 3495.89$ | 0.99999 | 0.22 | 0.65 |
| NPON | 0.78–50.00 | $y = 26{,}805.64x + 1579.48$ | 0.99999 | 0.05 | 0.15 |
| PON | 3.13–200.00 | $y = 13{,}843.50x + 4372.92$ | 0.99999 | 0.21 | 0.65 |
| API | 0.31–20.00 | $y = 39{,}370.97x + 466.23$ | 0.99999 | 0.02 | 0.05 |
| GLY | 1.56–100.00 | $y = 9792.06x + 907.65$ | 0.99999 | 0.08 | 0.26 |
| HON | 0.31–20.00 | $y = 18{,}705.23x + 920.39$ | 0.99999 | 0.02 | 0.05 |
| DEC | 0.78–50.00 | $y = 34{,}364.13x - 379.84$ | 0.99998 | 0.15 | 0.47 |
| DECA | 0.78–50.00 | $y = 22{,}150.45x - 68.44$ | 0.99999 | 0.15 | 0.47 |
| PRAA | 0.31–20.00 | $y = 17{,}562.36x + 2511.48$ | 0.99999 | 0.01 | 0.04 |

[a] $y$: peak area (mAU) of compounds; $x$: concentration (µg/mL) of compounds.

### 2.4. Preparations of Sample and Standard Stock Solutions for HPLC Analysis

In order to simultaneously determine the 14 marker components (LIQA, LIQ, NOD, NAR, NARG, HES, NPON, PON, API, GLY, HON, DEC, DECA, and PRAA) in the SDGT aqueous decoction, 10 mL of 70% methanol was added to 100 mg of the lyophilized SDGT sample, followed by ultrasonic extraction via a Branson 8510 (Denbury, CT, USA) ultra-sonicator. The extract was filtered through a 0.2 µm membrane filter (Pall Life Sciences, Ann Arbor, MI, USA) and injected into the HPLC system.

Standard stock solutions of the 14 reference standard compounds were prepared at a concentration of 1000.0 µg/mL using methanol whilst the prepared stock solutions were stored in a refrigerator until required.

### 2.5. HPLC–PDA Apparatus and Conditions for the Simultaneous Quantification

The simultaneous analysis for the quality assessment of SDGT was carried out with Shimadzu Prominence LC-20A series HPLC systems (Kyoto, Japan) consisting of two delivery systems (LC-20AT), an online degasser (DGU-20A$_3$), a forced air circulation type column oven (CTO-20A), an automatic sample injector (SIL-20A), a PDA (SPD-M20A), and LabSolution software (Version 5.53, SP3). A Phenomenex Gemini C$_{18}$ analytical reverse-phase column (4.6 × 250 mm, 5 µm; Torrance, CA, USA) maintained at 40 °C was used for the separation of all analytes. Mobile phases were (A) 0.1% (*v/v*) aqueous FA and (B) acetonitrile with 0.1% (*v/v*) FA with the following gradient conditions: start–40 min, 5–60% B; 40–50 min, 60–100% B; 50–55 min, 100% B; 55–60 min, 100–5% B. The flow rate of the mobile phase was 1.0 mL/min, and 10 µL of the sample solution was injected.

### 2.6. Validation of the HPLC Analytical Procedure

Generally, to apply the developed method for the simultaneous quantitative analysis and standardization of TKM formulations, the method must be validated. In this study, we validated the analytical procedure in accordance with the following strict parameters: limit of detection (LOD), limit of quantification (LOQ), and accuracy, and precision according to the International Conference on Harmonisation (ICH) guidelines [22].

In short, the linearity of these parameters was confirmed by the value of the coefficient determination ($r^2$) of the calibration curve, drawn at different concentration ranges of each marker

compound. LOD and LOQ concentrations were set using equations where $\sigma$ is the standard deviation of the *y*-intercept and *S* is the slope of the calibration curve.

$$\text{LOD} = 3.3 \times \frac{\sigma}{S}$$

The accuracy was evaluated based on the recovery test of the standard addition method, using the equation:

$$\text{Recovery (\%)} = \frac{(\text{found concentration} - \text{original concentration})}{\text{spiked concentration}} \times 100$$

where precision was verified by intraday and interday precision and repeatability measurements. Intraday and interday precisions were measured five times in a single day and on three consecutive days at different concentrations (low, medium, and high) using the standard solution. The relative standard deviation (RSD) of the recorded series of repeatability was also measured six times using the equation:

$$\text{RSD (\%)} = \frac{\text{standard deviation (SD)}}{\text{mean}} \times 100$$

The system suitability of the assay was evaluated using the parameters such as capacity factor (*k'*), selectivity factor (*α*), resolution (*Rs*), number of theoretical plates (*N*), and tailing factor (*Tf*).

### 2.7. LC–MS/MS Apparatus and Conditions for the Simultaneous Quantification

An ACQUITY UPLC H-Class (Waters, Milford, MA, USA) LC system coupled to TQ-S micro triple quadrupole mass spectrometer (Xevo, Milford, MA, USA) with an electrospray ionization (ESI) source was used for simultaneous analysis of the 14 marker components in SDGT. The ACQUITY UPLC BEH $C_{18}$ analytical column (2.1 × 100 mm, 1.7 μm) maintained at 40 °C was used for separation of the components. Mobile phases were (A) aqueous 0.1% (*v/v*) FA and (B) acetonitrile, with gradient elution: 10% B at start–1.0 min, 10–40% B at 1.0–6.0 min, 40–95% B at 6.0–10.0 min, 95% B at 10.0–15.0 min, 95–10% B at 15.0–15.1 min, and 10% B at 15.1–18.0 min. The flow rate for separation and detection was 0.3 mL/min and the injection volume was 2.0 μL. The temperature of the prepared sample was maintained at 5 °C. The 14 marker components were detected in positive and negative ion modes of the LC–MS/MS multiple reaction monitoring (MRM) mode. The optimized LC–MS/MS MRM parameters (collision energy, cone voltage, and transition) are presented in Table S2. Other parameters were: capillary voltage, 1.2 kV; source temperature, 150 °C; desolvation temperature, 450 °C; desolvation gas flow, 800 L/h; cone gas flow, 50 L/h. All data were processed using Waters MassLynx software (Version 4.2; Waters, Milford, MA, USA).

## 3. Results and Discussion

### 3.1. Optimization of HPLC Chromatographic Separation Conditions

Appropriate conditions were established with respect to the acidic water–acetonitrile mobile phase, SunFire reverse-phase column, and the column temperature of 40 °C, among the tested conditions for the simultaneous analysis of the 14 marker components (LIQA, LIQ, NOD, NAR, NARG, HES, NPON, PON, API, GLY, HON, DEC, DECA, and PRAA) of SDGT. Through the use of these optimum conditions, the 14 marker components were efficiently separated within 50 min with a resolution > 1.4. Retention times of the marker components were 19.02, 19.40, 20.31, 20.52, 21.23, 21.67, 26.13, 26.77, 30.66, 36.05, 44.99, 46.31, 46.61, and 47.42 min, respectively (Figure 1).

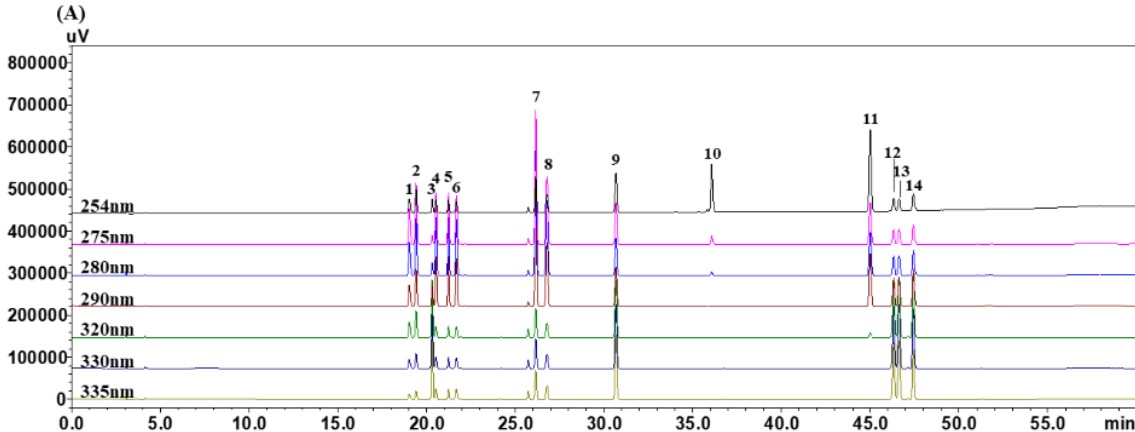

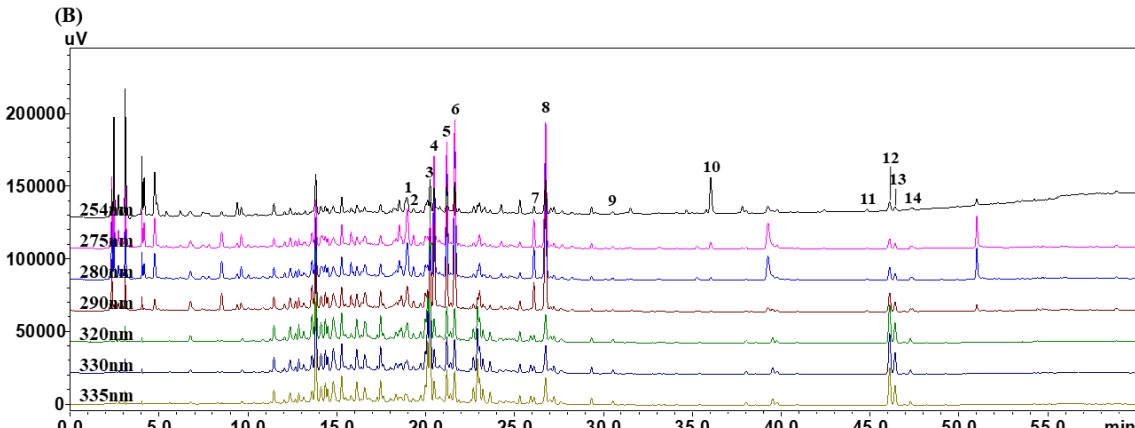

**Figure 1.** HPLC chromatograms of (**A**) standard solution and (**B**) Sojadodamgangki-tang (SDGT) sample. (1) liquiritin apioside (LIQA), (2) liquiritin (LIQ), (3) nodakenin (NOD), (4) narirutin (NAR), (5) naringin (NARG), (6) hesperidin (HES), (7) neoponcirin (NPON), (8) poncirin (PON), (9) apigenin (API), (10) glycyrrhizin (GLY), (11) honokiol (HON), (12) decursin (DEC), (13) decursinol angelate (DECA), and (14) praeruptorin A (PRAA).

### 3.2. Method Validation of the HPLC Analytical Method

System characteristics such as capacity factor (*k*), selectivity (*α*), theoretical plate number (*N*), resolution (*Rs*), and tailing factor (*Tf*), were all evaluated to ensure that the complete system consisting of the analytical instrument, analytical operation, and samples to be analyzed was performing as required. The results (Table S3) confirmed that all parameters were suitable for the method. As shown in Table 1, the $r^2$ values of all the components showed excellent linearity of ≥0.99996, and LOD and LOQ concentrations were calculated to be 0.01–0.37 μg/mL and 0.04–1.11 μg/mL, respectively. The extraction recovery (%) for the evaluation of each marker component's accuracy at three different concentrations was determined to be 95.89–104.18% and the RSD value was calculated to be less than 2.50% (Table 2). Intraday, interday, and repeatability precisions showed an RSD (%) of less than 3.00% (Table 3). This result confirmed the validity of the established simultaneous analysis method for the quality assessment of SDGT.

**Table 2.** Recovery test for the assay of 14 components in SDGT using HPLC–PDA.

| Compound | Original Conc. (μg/mL) | Spiked Conc. (μg/mL) | Found Conc. (μg/mL) | Recovery (%) | SD | RSD (%) |
|---|---|---|---|---|---|---|
| LIQA | 20.74 | 4.00 | 24.73 | 99.78 | 1.17 | 1.18 |
| | | 10.00 | 30.58 | 98.44 | 0.58 | 0.59 |
| | | 20.00 | 40.40 | 98.33 | 0.73 | 0.74 |
| LIQ | 4.14 | 1.00 | 5.12 | 97.93 | 1.68 | 1.72 |
| | | 2.00 | 6.14 | 99.63 | 0.99 | 1.00 |
| | | 4.00 | 8.03 | 97.14 | 0.57 | 0.58 |
| NOD | 20.56 | 4.00 | 24.46 | 97.46 | 0.65 | 0.67 |
| | | 10.00 | 30.65 | 100.96 | 1.42 | 1.41 |
| | | 20.00 | 41.39 | 104.18 | 0.52 | 0.50 |
| NAR | 21.52 | 4.00 | 25.46 | 98.47 | 1.28 | 1.30 |
| | | 10.00 | 32.22 | 100.91 | 0.54 | 0.54 |
| | | 20.00 | 41.37 | 99.23 | 1.44 | 1.45 |
| NARG | 31.04 | 6.00 | 36.97 | 98.92 | 2.42 | 2.45 |
| | | 15.00 | 46.45 | 102.76 | 1.32 | 1.29 |
| | | 30.00 | 61.25 | 100.71 | 0.79 | 0.78 |
| HES | 39.50 | 8.00 | 47.41 | 98.89 | 2.04 | 2.06 |
| | | 20.00 | 59.42 | 99.62 | 2.11 | 2.12 |
| | | 40.00 | 79.70 | 100.50 | 0.64 | 0.64 |
| NPON | 5.75 | 1.00 | 6.77 | 101.80 | 2.21 | 2.17 |
| | | 2.00 | 7.73 | 99.08 | 1.91 | 1.92 |
| | | 4.00 | 9.79 | 100.88 | 0.66 | 0.66 |
| PON | 53.59 | 10.00 | 63.43 | 98.39 | 1.31 | 1.33 |
| | | 25.00 | 78.74 | 100.60 | 0.99 | 0.99 |
| | | 50.00 | 105.46 | 103.75 | 0.77 | 0.75 |
| API | 0.57 | 1.00 | 1.56 | 98.94 | 0.86 | 0.87 |
| | | 2.00 | 2.52 | 97.72 | 1.60 | 1.64 |
| | | 4.00 | 4.43 | 96.49 | 1.00 | 1.04 |
| GLY | 20.56 | 4.00 | 24.47 | 97.59 | 1.63 | 1.68 |
| | | 10.00 | 30.34 | 97.71 | 1.05 | 1.07 |
| | | 20.00 | 39.74 | 95.89 | 0.61 | 0.64 |
| HON | 0.47 | 1.00 | 1.51 | 103.56 | 0.83 | 0.80 |
| | | 2.00 | 2.43 | 97.60 | 1.00 | 1.03 |
| | | 4.00 | 4.57 | 102.39 | 0.24 | 0.23 |
| DEC | 6.50 | 1.00 | 7.52 | 101.99 | 2.64 | 2.59 |
| | | 2.00 | 8.45 | 97.16 | 1.06 | 1.09 |
| | | 4.00 | 10.36 | 96.30 | 0.26 | 0.27 |
| DECA | 5.19 | 1.00 | 6.22 | 102.69 | 1.26 | 1.22 |
| | | 2.00 | 7.18 | 99.47 | 1.67 | 1.68 |
| | | 4.00 | 9.12 | 98.11 | 0.72 | 0.73 |
| PRAA | 1.31 | 1.00 | 2.33 | 102.09 | 1.72 | 1.68 |
| | | 2.00 | 3.27 | 98.14 | 2.39 | 2.44 |
| | | 4.00 | 5.36 | 101.43 | 0.52 | 0.51 |

**Table 3.** Precision assay for the 14 marker compounds in SDGT using HPLC–PDA.

| Analyte | Conc. (µg/mL) | Intraday (*n* = 5) | | | Interday (*n* = 5) | | | Repeatability (*n* = 6) | |
|---|---|---|---|---|---|---|---|---|---|
| | | Measured Conc. (µg/mL) | Precision (%) | Accuracy (%) | Measured Conc. (µg/mL) | Precision (%) | Accuracy (%) | RSD (%) of Retention Time | RSD (%) of Peak Area |
| LIQA | 25.00 | 25.38 | 0.45 | 101.52 | 25.89 | 1.61 | 103.56 | | |
| | 50.00 | 50.59 | 0.54 | 101.19 | 51.01 | 1.23 | 102.02 | 0.05 | 0.68 |
| | 100.00 | 99.25 | 0.35 | 99.25 | 100.11 | 1.41 | 100.11 | | |
| LIQ | 12.50 | 12.66 | 0.39 | 101.31 | 12.93 | 1.68 | 103.41 | | |
| | 20.00 | 25.24 | 0.50 | 100.97 | 25.45 | 1.30 | 101.79 | 0.04 | 0.65 |
| | 50.00 | 49.55 | 0.34 | 99.10 | 49.99 | 1.50 | 99.99 | | |
| NOD | 25.00 | 25.00 | 0.86 | 100.00 | 24.56 | 2.17 | 98.25 | | |
| | 50.00 | 50.63 | 1.00 | 101.25 | 50.74 | 1.00 | 101.48 | 0.06 | 0.65 |
| | 100.00 | 100.57 | 0.34 | 100.57 | 97.58 | 2.71 | 97.58 | | |
| NAR | 25.00 | 25.47 | 0.46 | 101.89 | 25.96 | 1.50 | 103.83 | | |
| | 50.00 | 50.78 | 0.56 | 101.56 | 51.14 | 1.15 | 102.27 | 0.06 | 0.64 |
| | 100.00 | 99.43 | 0.37 | 99.43 | 100.23 | 1.29 | 100.23 | | |
| NARG | 37.50 | 38.03 | 0.42 | 101.41 | 38.80 | 1.63 | 103.47 | | |
| | 75.00 | 75.87 | 0.50 | 101.16 | 76.47 | 1.21 | 101.96 | 0.05 | 0.62 |
| | 150.00 | 148.46 | 0.37 | 98.97 | 149.78 | 1.41 | 99.85 | | |
| HES | 37.50 | 37.92 | 0.35 | 101.11 | 38.84 | 1.86 | 103.57 | | |
| | 75.00 | 75.46 | 0.50 | 100.61 | 76.46 | 1.45 | 101.95 | 0.04 | 0.58 |
| | 150.00 | 147.67 | 0.39 | 98.45 | 149.66 | 1.61 | 99.77 | | |
| NPON | 12.50 | 12.70 | 0.42 | 101.58 | 12.96 | 1.64 | 103.69 | | |
| | 25.00 | 25.33 | 0.40 | 101.33 | 25.54 | 1.22 | 102.17 | 0.05 | 0.67 |
| | 50.00 | 49.71 | 0.29 | 99.42 | 50.11 | 1.38 | 100.22 | | |
| PON | 50.00 | 50.76 | 0.42 | 101.53 | 51.87 | 1.72 | 103.73 | | |
| | 100.00 | 101.41 | 0.42 | 101.41 | 102.21 | 1.21 | 102.21 | 0.05 | 0.67 |
| | 200.00 | 199.04 | 0.28 | 99.52 | 200.47 | 1.35 | 100.23 | | |
| API | 5.00 | 5.03 | 0.57 | 100.51 | 5.14 | 1.89 | 102.86 | | |
| | 10.00 | 10.07 | 0.57 | 100.68 | 10.16 | 1.29 | 101.57 | 0.03 | 0.61 |
| | 20.00 | 19.82 | 0.34 | 99.10 | 20.01 | 1.46 | 100.04 | | |
| GLY | 25.00 | 25.57 | 1.28 | 102.26 | 25.96 | 1.41 | 103.85 | | |
| | 50.00 | 50.90 | 0.20 | 101.80 | 51.27 | 1.68 | 102.53 | 0.03 | 0.68 |
| | 100.00 | 99.68 | 0.27 | 99.68 | 100.11 | 1.29 | 100.11 | | |
| HON | 5.00 | 5.05 | 0.33 | 100.95 | 5.16 | 1.79 | 103.26 | | |
| | 10.00 | 10.11 | 0.58 | 101.08 | 10.18 | 1.21 | 101.83 | 0.02 | 0.60 |
| | 20.00 | 19.84 | 0.38 | 99.20 | 20.02 | 1.37 | 100.08 | | |
| DEC | 12.50 | 12.53 | 0.73 | 100.24 | 12.34 | 1.90 | 98.73 | | |
| | 20.00 | 25.35 | 1.03 | 101.42 | 25.43 | 1.00 | 101.74 | 0.02 | 0.60 |
| | 50.00 | 50.21 | 0.24 | 100.42 | 48.85 | 2.51 | 97.71 | | |
| DECA | 12.50 | 12.54 | 0.80 | 100.33 | 12.37 | 1.89 | 98.95 | | |
| | 20.00 | 25.38 | 1.01 | 101.51 | 25.47 | 1.01 | 101.87 | 0.02 | 0.58 |
| | 50.00 | 50.31 | 0.34 | 100.63 | 48.98 | 2.47 | 97.96 | | |
| PRAA | 5.00 | 5.04 | 0.68 | 100.80 | 5.15 | 1.80 | 103.08 | | |
| | 10.00 | 10.10 | 0.54 | 100.95 | 10.17 | 1.30 | 101.74 | 0.02 | 0.60 |
| | 20.00 | 19.83 | 0.41 | 99.15 | 20.02 | 1.43 | 100.11 | | |

### 3.3. LC–MS/MS Confirmation

As shown in Table S2, all marker components in the LC–MS system were identified from their molecular ion peaks in the positive or negative ion modes. Six analytes (LIQA, LIQ, NAR, NARG, GLY, and HON) were detected using the negative ion mode [M – H]⁻ at *m/z* 549.4, 417.3, 579.5, 579.5, 821.6, and 265.1, respectively. Seven analytes (NOD, HES, NPON, PON, API, DEC, and DECA) were also detected using the positive ion mode [M + H]⁺ at *m/z* 409.2, 611.4, 595.1, 595.1, 271.2, 329.2, and 329.2, respectively. In the case of PRAA, the molecular ion peak was confirmed at *m/z* 404.1 in the form of [M + H₂O]⁺ in the positive ion mode. The $r^2$ of the prepared calibration curve for each analyte was more than 0.99, demonstrating excellent linearity at the tested concentration ranges (Table S4). The LOD and LOQ values of each component were 0.001–0.710 ng/mL and 0.003–2.131 ng/mL, respectively, based on signal-to-noise ratios of 3.3 and 10. (Table S4).

### 3.4. Simultaneous Quantification of the 14 Marker Components in Freeze-Dried SDGT Using HPLC–PDA

The established and validated HPLC assay was then applied to the simultaneous analysis of the 14 marker components (LIQA, LIQ, NOD, NAR, NARG, HES, NPON, PON, API, GLY, HON, DEC, DECA, and PRAA) for the quality control of SDGT. Each sample was analyzed in triplicate across three batches. Quantification of each marker component was conducted with a PDA detector, which was then scanned simultaneously in the range of 190–400 nm; the detection wavelengths for quantification

were 254 nm (GLY), 275 nm (LIQA and LIQ), 280 nm (NAR, NARG, HES, NPON, and PON), 290 nm (HON), 320 nm (PRAA), 330 nm (DEC and DECA), and 335 nm (NOD and API). The concentrations of the 14 markers were found to range from 0.05 mg/g to 5.25 mg/g (Table 4). Of these markers, PON, the marker component of Ponciri Fructus Immaturus, was found to be the most abundant in SDGT (5.20–5.25 mg/g).

**Table 4.** Amount of the 14 marker components in SDGT based on HPLC–PDA analysis (*n* = 3).

| Compound | Batch No. | | | | | |
| | **1** | | **2** | | **3** | |
| | **Mean (mg/g) ± SD (×10⁻²)** | **RSD (%)** | **Mean (mg/g) ± SD (×10⁻²)** | **RSD (%)** | **Mean (mg/g) ± SD (×10⁻²)** | **RSD (%)** |
|---|---|---|---|---|---|---|
| LIQA | 2.07 ± 2.77 | 1.34 | 2.08 ± 5.30 | 2.54 | 2.05 ± 1.49 | 0.73 |
| LIQ | 0.41 ± 1.15 | 2.80 | 0.41 ± 0.07 | 0.18 | 0.42 ± 1.15 | 2.72 |
| NOD | 2.09 ± 1.43 | 0.68 | 2.08 ± 2.41 | 1.16 | 2.13 ± 1.87 | 0.88 |
| NAR | 2.20 ± 1.94 | 0.88 | 2.23 ± 0.63 | 0.28 | 2.23 ± 2.80 | 1.25 |
| NARG | 3.02 ± 4.10 | 1.36 | 3.05 ± 1.45 | 0.48 | 3.05 ± 2.31 | 0.76 |
| HES | 3.82 ± 8.20 | 2.14 | 3.82 ± 3.09 | 0.81 | 3.86 ± 1.70 | 0.44 |
| NPON | 0.55 ± 0.57 | 1.05 | 0.56 ± 1.17 | 2.10 | 0.56 ± 0.24 | 0.43 |
| PON | 5.24 ± 11.53 | 2.20 | 5.20 ± 0.57 | 0.11 | 5.25 ± 1.92 | 0.37 |
| API | 0.05 ± 0.02 | 0.37 | 0.05 ± 0.02 | 0.36 | 0.06 ± 0.05 | 0.83 |
| GLY | 1.98 ± 1.19 | 0.60 | 1.99 ± 2.08 | 1.05 | 2.02 ± 0.72 | 0.36 |
| HON | 0.05 ± 0.04 | 0.77 | 0.05 ± 0.08 | 1.58 | 0.05 ± 0.07 | 1.34 |
| DEC | 0.66 ± 0.11 | 0.17 | 0.53 ± 0.06 | 0.11 | 0.66 ± 0.08 | 0.12 |
| DECA | 0.53 ± 0.14 | 0.27 | 0.66 ± 0.17 | 0.26 | 0.53 ± 0.06 | 0.11 |
| PRAA | 0.12 ± 0.12 | 0.93 | 0.13 ± 0.05 | 0.37 | 0.13 ± 0.06 | 0.48 |

*3.5. Simultaneous Determination of the 14 Analytes in Freeze-Dried SDGT Samples Using LC–MS/MS*

The 14 marker components in the SDGT extract were simultaneously analyzed and quantified by using the established LC–MS/MS MRM assay. The 14 analytes (LIQA, LIQ, NOD, NAR, NARG, HES, NPON, PON, API, GLY, HON, DEC, DECA, and PRAA) for simultaneous determination in the sample were detected at 4.65, 4.70, 5.03, 5.12, 5.27, 5.45, 6.58, 6.58, 7.15, 8.09, 9.74, 9.91, 9.96, and 10.14 min (Figure 2 and Figure S2), respectively, and the concentrations of these analytes were found to be 0.005–0.384 mg/g (Table 5).

**Table 5.** Amount of the 14 marker components in SDGT based on LC–MS/MS analysis (*n* = 3).

| Compound | Amount (mg/g) | | |
| | **Mean** | **SD (×10⁻²)** | **RSD (%)** |
|---|---|---|---|
| LIQA | 0.171 | 0.930 | 5.448 |
| LIQ | 0.037 | 0.029 | 0.785 |
| NOD | 0.309 | 2.034 | 6.592 |
| NAR | 0.156 | 1.161 | 7.428 |
| NARG | 0.246 | 2.382 | 9.691 |
| HES | 0.384 | 1.910 | 4.971 |
| NPON+PON | 0.046 | 0.223 | 4.841 |
| API | 0.005 | 0.002 | 0.495 |
| GLY | 0.206 | 1.188 | 5.763 |
| HON | 0.005 | 0.013 | 2.289 |
| DEC | 0.138 | 0.943 | 6.837 |
| DECA | 0.142 | 0.938 | 6.588 |
| PRAA | 0.029 | 0.064 | 2.208 |

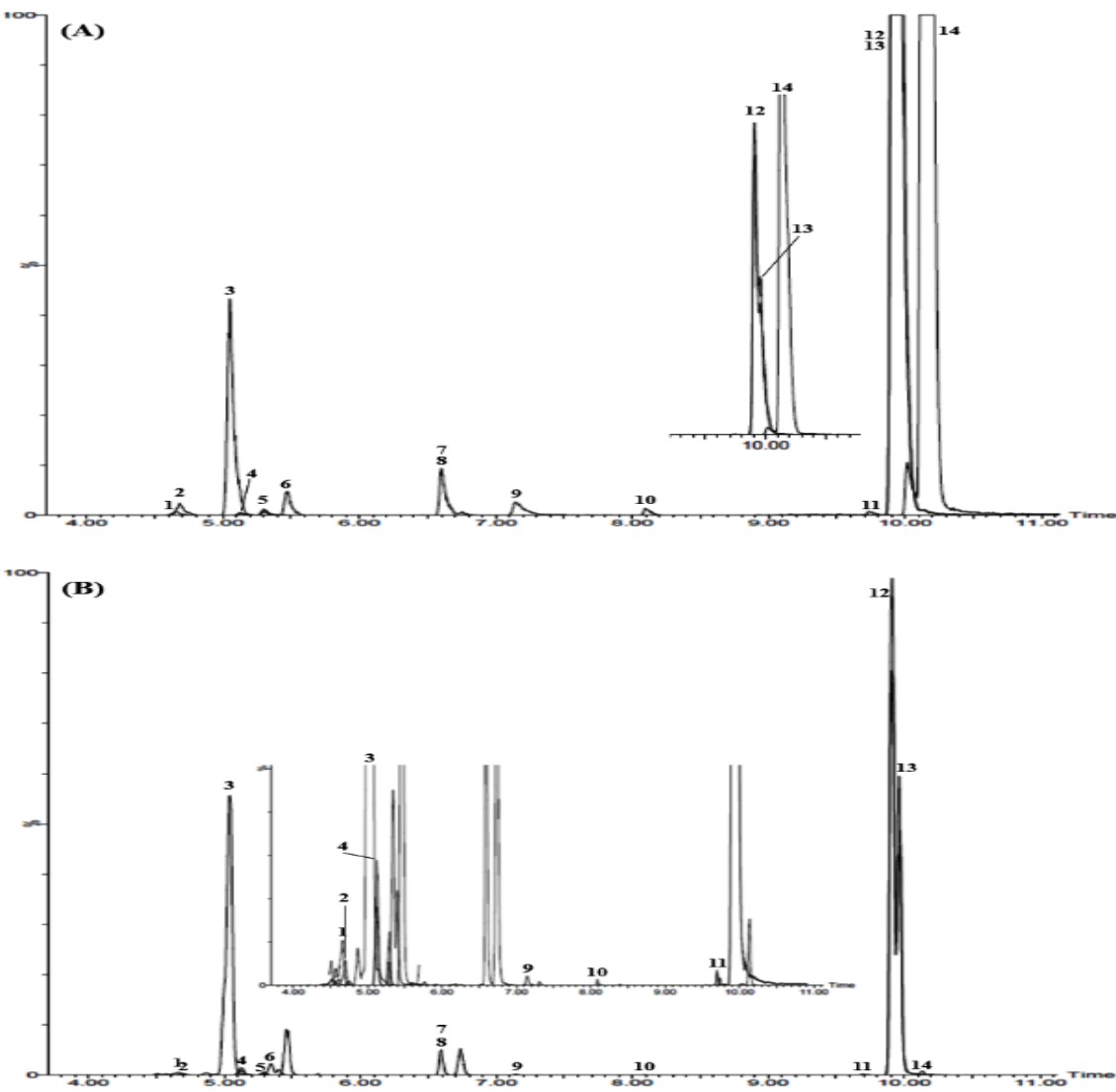

**Figure 2.** Total ion chromatograms of (**A**) the 14 reference standard solutions and (**B**) SDGT extract by LC–MS/MS MRM mode. (1) LIQA, (2) LIQ, (3) NOD, (4) NAR, (5) NARG, (6) HES, (7) NPON, (8) PON, (9) API, (10) GLY, (11) HON, (12) DEC, (13) DECA, and (14) PRAA.

## 4. Conclusions

In this study, both qualitative and quantitative analysis for the quality control of SDGT using HPLC–PDA and LC–MS/MS were developed and validated for the first time. Simultaneous analysis of the 14 marker components in SDGT via the use of the two analytical methods was also successfully performed and validated. These newly validated and established HPLC–PDA and LC–MS/MS methods are expected to be effective protocols for the quality control of SDGT and related herbal formulas going forward.

**Supplementary Materials:** The following are available online at http://www.mdpi.com/2076-3417/10/8/2804/s1, Figure S1: Chemical structures of the 14 marker components in Sojadodamgangki-tang (SDGT); Figure S2: Extracted ion chromatograms of (A) the reference standard and (B) SDGT extract by LC–MS/MS MRM mode. (1) LIQA, (2) LIQ, (3) NOD, (4) NAR, (5) NARG, (6) HES, (7) NPON, (8) PON, (9) API, (10) GLY, (11) HON, (12) DEC, (13) DECA, and (14) PRAA. Table S1: Composition of SDGT; Table S2: MRM parameters for LC–MS/MS analysis of the 14 marker components in SDGT; Table S3: System suitability for the 14 marker compounds using HPLC–PDA; Table S4: The linear range, regression equation, $r^2$, LOD, and LOQ of the analytes from SDGT using LC–MS/MS.

**Author Contributions:** Conceptualization, C.-S.S. and M.-Y.L.; performing experiments and analyzing data, C.-S.S.; writing—original draft preparation, C.-S.S.; funding acquisition, M.-Y.L. All authors have read and agreed to the published version of the manuscript.

**Funding:** This research was supported by a grant from the Korea Institute of Oriental Medicine (grant number KSN1812230).

**Conflicts of Interest:** The authors have declared no conflict of interest.

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
