# Peer review of "HPLC–PDA and LC–MS/MS Analysis for the Simultaneous Quantification of the 14 Marker Components in Sojadodamgangki-Tang"

_applsci, doi:10.3390/app10082804_

Round 1

Reviewer 1 Report

This study described the LC-PDA and LC-MS/MS method development for the analysis of 14 marker components for quality control of SDGT. The study is well-organized, but no optimization process is provided in the result part. Only final results had been showed in the manuscript. It would be better if more discussion and explanation could be added in the result part. Many errors are found in the tables, please modify these errors. This manuscript could be accepted only after major revision.

Other points:

According to LC-PDA, PON and NPON could be fully separated by C18 column. For LC-MS/MS, although they share the same transition on MS/MS, it still could be separated by the same C18 column. Please discuss the reason for changing column and why not to separate them in this study? For confirmation purpose, I think it would be better to separate them.

Table S2. The MW column showed the molecular weight, but the unit should not be m/z.

The values and units of mean ± SD showed in table 4 are confusing. For example, “0.41 ± 1.15”, does it means “0.41 ± 0.115”?

Table 4 showed the amount of the 14 marker components in SDGT with the values of Mean ± SD. But most of the SD values are larger than mean values, that indicate the repeatability of this batch test are not good enough. It would be calculation errors…For example, “0.41 ± 0.115” have the RSD of 28.0% not 2.8%.

Table 5 showed the mean, SD, and RSD, but according to the values, I can’t figure out how to calculate them. Usually, it would be SD/mean = RSD. For the LIQA compound, the SD is 0.93, and the mean is 0.171, so the RSD (%) should be 543.8%! That’s not good for a common analysis…please check all the numbers again.

Figure 2. It would be better to provide the extracted ion chromatograph for individual components instead of total ion chromatogram as the demonstration.

Author Response

Comments and Suggestions for Authors

This study described the LC-PDA and LC-MS/MS method development for the analysis of 14 marker components for quality control of SDGT. The study is well-organized, but no optimization process is provided in the result part. Only final results had been showed in the manuscript. It would be better if more discussion and explanation could be added in the result part. Many errors are found in the tables, please modify these errors. This manuscript could be accepted only after major revision.

Other points:

1. According to LC-PDA, PON and NPON could be fully separated by C18 column. For LC-MS/MS, although they share the same transition on MS/MS, it still could be separated by the same C18 column. Please discuss the reason for changing column and why not to separate them in this study? For confirmation purpose, I think it would be better to separate them.

Answer) We appreciate your comment. As reviewer’s comment, we are unfortunately unable to separate PON and NPON due to the same transition (precursor ion to product ion) of two compounds under these LC-MS/MS analytical condition (see Fig. S2). Only this is considered a weakness of this study, and we will ensure that all components are separated in future studies.

2. Table S2. The MW column showed the molecular weight, but the unit should not be m/z.

Answer) We appreciate your comment. In Table S2, we removed the m/z from the MW column.

3. The values and units of mean ± SD showed in table 4 are confusing. For example, “0.41 ± 1.15”, does it means “0.41 ± 0.115”?

Answer) We appreciate your comment. The reviewer’s opinion is correct. However, in SD expression, it is 10-2 not 10-1. For example, “0.41 ± 1.15” means “0.41 ± 0.0115”.

4. Table 4 showed the amount of the 14 marker components in SDGT with the values of Mean ± SD. But most of the SD values are larger than mean values, that indicate the repeatability of this batch test are not good enough. It would be calculation errors…For example, “0.41 ± 0.115” have the RSD of 28.0% not 2.8%.

Answer) We appreciate your comment. It is calculation error, as reviewer’s opinion. SD value is not presented correctly. It is 10-2 not 10-1.

5. Table 5 showed the mean, SD, and RSD, but according to the values, I can’t figure out how to calculate them. Usually, it would be SD/mean = RSD. For the LIQA compound, the SD is 0.93, and the mean is 0.171, so the RSD (%) should be 543.8%! That’s not good for a common analysis…please check all the numbers again.

Answer) We appreciate your comment. It is calculation error. SD value was corrected to 10-2.

6. Figure 2. It would be better to provide the extracted ion chromatograph for individual components instead of total ion chromatogram as the demonstration.

Answer) We appreciate your comment. We added the extracted ion chromatograms of the components in the standard and sample to Supplementary materials (see Fig. S2).

Reviewer 2 Report

The authors present an HPLC-UV characterization method for the Korean medicine Sojadodamgangki-tang which is validated by mass spectrometric analysis. While the manuscript is not groundbreaking, it could be of potential interest for the readers of applied sciences.

Please find below some minor comments.

The aim of the study and the choice of the specific target compounds is not supported in a convincing way by the manuscript. Are they chosen due to their abundancy, to their biological activity etc?

Furthermore their cas numbers, or another identifier should be presented together to their chemical structure since they are isomeric forms of them and it’s not clear what has been used.

Furthermore, in table 1, the linearity r2 should not be presented rounded to 1.

Author Response

Comments and Suggestions for Authors

The authors present an HPLC-UV characterization method for the Korean medicine Sojadodamgangki-tang which is validated by mass spectrometric analysis. While the manuscript is not groundbreaking, it could be of potential interest for the readers of applied sciences.

Please find below some minor comments.

1. The aim of the study and the choice of the specific target compounds is not supported in a convincing way by the manuscript. Are they chosen due to their abundancy, to their biological activity etc?

Answer) We appreciate your comment. This study focuses on the simultaneous analysis for quality control of Sojadodamgangki-tang. The selected 14 marker components were primarily targeted for abundance among each herbal medicine composed of Sojadodamgangki-tang.

2. Furthermore their cas numbers, or another identifier should be presented together to their chemical structure since they are isomeric forms of them and it’s not clear what has been used.

Answer) We appreciate your comment. We added the CAS number of each marker component in section 2.2 Chemicas and reagents.

3. Furthermore, in table 1, the linearity r2 should not be presented rounded to 1.

Answer) We appreciate your comment. In Table 1, the linearity (r2) is presented to 5th decimal place.

Round 2

Reviewer 1 Report

New provided EIC showed several news issues:

1.The EIC of LIQA and LIQ showed they were seriously affect by other compounds in the SDGT extract. It may lead to quantification bias.

2.Any reason for explanation of the DEC and DECA have similar EIC? According to Table S2, DEC have RT of 9.91, while DECA have RT of 9.96.They are not fully separated! It seems quantification of DEC and DECA will have very big problem. 

Author Response

1.The EIC of LIQA and LIQ showed they were seriously affect by other compounds in the SDGT extract. It may lead to quantification bias.

 Answer) We appreciate your comment. We modified the EIC of LIQA and LIQ in the SDGT extract (see Fig. S2).

2.Any reason for explanation of the DEC and DECA have similar EIC? According to Table S2, DEC have RT of 9.91, while DECA have RT of 9.96. They are not fully separated! It seems quantification of DEC and DECA will have very big problem. 

 Answer) We appreciate your comment. DEC and DECA are structural isomers with the same fragmentation pattern in MS. Therefore, the two components were separated and quantified by the difference of RT. As reviewer’s comment, although two compounds were not completely separated, quantification showed possible resolution. The following reference [1] show LC-MS/MS chromatogram similar to our result.

1. Kim, S.J.; Ko, S.M.; Choi, E.J.; Ham, S.H.; Kwon, Y.D.; Lee, Y.B.; Cho, H.Y. Simultaneous determination of decursin, decursinol angelate, nodakenin, and decursinol of Angelica gigas Nakai in human plasma by UHPLC-MS/MS: Aplication to pharmacokinetic study. Molecules 2018, 23, 1019. Doi:10.3390/molecules23051019.

Round 3

Reviewer 1 Report

Although I am not satisfied with the LC-MS data, this paper could be accepted in the present form.